# Neural dynamics between anterior insular cortex and right supramarginal gyrus dissociate genuine affect sharing from perceptual saliency of pretended pain

Yili Zhao[1], Lei Zhang[1], Markus Rütgen[1,2], Ronald Sladky[1], Claus Lamm[1,2]*

[1]Social, Cognitive and Affective Neuroscience Unit, Department of Cognition, Emotion, and Methods in Psychology, Faculty of Psychology, University of Vienna, Vienna, Austria; [2]Vienna Cognitive Science Hub, University of Vienna, Vienna, Austria

**Abstract** Empathy for pain engages both shared affective responses and self-other distinction. In this study, we addressed the highly debated question of whether neural responses previously linked to affect sharing could result from the perception of salient affective displays. Moreover, we investigated how the brain network involved in affect sharing and self-other distinction underpinned our response to a pain that is either perceived as genuine or pretended (while in fact both were acted for reasons of experimental control). We found stronger activations in regions associated with affect sharing (anterior insula [aIns] and anterior mid-cingulate cortex) as well as with affective self-other distinction (right supramarginal gyrus [rSMG]), in participants watching video clips of genuine vs. pretended facial expressions of pain. Using dynamic causal modeling, we then assessed the neural dynamics between the right aIns and rSMG in these two conditions. This revealed a reduced inhibitory effect on the aIns to rSMG connection for genuine pain compared to pretended pain. For genuine pain only, brain-to-behavior regression analyses highlighted a linkage between this inhibitory effect on the one hand, and pain ratings as well as empathic traits on the other. These findings imply that if the pain of others is genuine and thus calls for an appropriate empathic response, neural responses in the aIns indeed seem related to affect sharing and self-other distinction is engaged to avoid empathic over-arousal. In contrast, if others merely pretend to be in pain, the perceptual salience of their painful expression results in neural responses that are down-regulated to avoid inappropriate affect sharing and social support.

*For correspondence:
claus.lamm@univie.ac.at

Competing interest: The authors declare that no competing interests exist.

## Introduction

As social beings, our own affective states are influenced by other people's feelings and affective states. The facial expression of pain by others acts as a distinctive cue to signal their pain to others, and thus results in sizeable affective responses in the observer. Certifying such responses as evidence for empathy, however, requires successful self-other distinction, the ability to distinguish the affective response experienced by ourselves from the affect experienced by the other person.

Studies using a wide variety of methods convergently have shown that observing others in pain engages neural responses aligning with those coding for the affective component of self-experienced pain, with the anterior insula (aIns) and the anterior mid-cingulate cortex (aMCC) being two key areas in which such an alignment has been detected (*Lamm et al., 2011*; *Rütgen et al., 2015*; *Jauniaux et al., 2019*; *Xiong et al., 2019*; *Zhou et al., 2020*; *Fallon et al., 2020*, for meta-analyses). However, there is consistent debate on whether activity observed in these areas should indeed be related to the sharing of pain affect or whether it may not rather result from automatic responses to salient

**eLife digest** Empathy enables us to share and understand the emotional states of other people, often based on their facial expressions. This empathic response involves being able to distinguish our own emotional state from someone else's, and it is influenced by how we recognize that person's emotion. In real life, knowing and identifying whether the facial expression we are witnessing reflects genuine or pretended pain is particularly important so that we can appropriately react to someone's emotions and avoid unnecessary personal distress.

How our brains manage to do this is still heavily debated. Two areas, the anterior insular (aIns for short) and the mid-cingulate cortex, appear to be activated when someone 'feels' someone else's pain. However, these regions might just automatically be triggered by vivid emotional facial expressions, regardless of whether we really respond to that pain.

To examine this question, Zhao et al. measured brain activity as healthy adults watched video clips of people either feeling or pretending to feel pain. The activation of aIns was particularly related to the emotional component that someone shared with another person's genuine pain, but not to pretended pain. This suggests that neurons in the aIns track a truly empathic response when seeing someone who is actually experiencing pain.

Effective connectivity analyses which reflect how brain areas 'crosstalk' also revealed distinct patterns when people viewed expressions of genuine, as opposed to pretended pain. Zhao et al. focused on the interactions between the aIns and the right supramarginal gyrus, a brain region which helps to distinguish another person's emotions from our own. This crosstalk tracked others' feelings when participants viewed expressions of genuine but not of pretended pain.

Put together, these findings provide a more refined model of empathy and its neural underpinnings. This will help further our understanding of conditions such as autism or depression, in which a person's social skills and emotional processing are impaired.

perceptual cues – with pain vividly expressed on the face being one particularly prominent example (*Zaki et al., 2016*, for review). It was thus one major aim of our study to address this question. In this respect, contextual factors, individuals' appraisals, and attentional processes would all impact their exact response to the affective states of others (*Gu and Han, 2007*; *Hein and Singer, 2008*, for review; *Lamm et al., 2010*; *Forbes and Hamilton, 2020*; *Zhao et al., 2021*). Recently, *Coll et al., 2017* have thus proposed a framework that attempts to capture these influences on affect sharing and empathic responses. This model posits that individuals who see identical negative facial expressions of others may have different empathic responses due to distinct contextual information and that this may depend on identification of the underlying affective state displayed by the other. In the current functional magnetic resonance imaging (fMRI) study, we therefore created a situation where we varied the genuineness of the pain affect felt by participants while keeping the perceptual saliency (i.e., the quality and strength of pain expressions) identical. To this end, participants were shown video clips of other persons who supposedly displayed genuine pain on their face vs. merely pretended to be in pain. Note that for reasons of experimental control, all painful expressions on the videos had been acted. This enabled us to interpret possible differences between conditions to the observers' appraisal of the situation rather than to putative visual and expressive differences. This way, we sought to identify the extent to which responses in affective nodes (such as the aIns and the aMCC) genuinely track the pain of others, rather than resulting predominantly from the salient facial expressions associated with the pain.

Another major aim of our study was to assess how self-other distinction allowed individuals to distinguish between the sharing of actual pain vs. regulating an inappropriate and potentially misleading 'sharing' of what in reality is only a pretended affective state. We focused on the right supramarginal gyrus (rSMG), which has been suggested to act as a major hub selectively engaged in *affective* self-other distinction (*Silani et al., 2013*; *Steinbeis et al., 2015*; *Hoffmann et al., 2016*; *Bukowski et al., 2020*). Though previous studies have indicated that rSMG is functionally connected with areas associated with affect processing (*Mars et al., 2012*; *Bukowski et al., 2020*), we lack more nuanced insights into how exactly rSMG interacts with these areas, and thus how it supports accurate empathic responses. Hence, we used dynamic causal modeling (DCM) to investigate the hypothesized brain

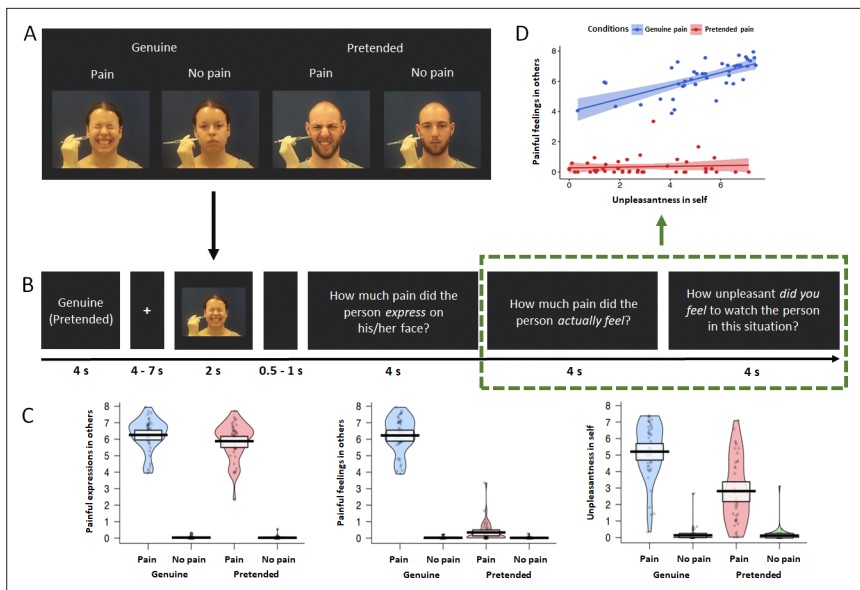

**Figure 1.** fMRI experimental design and behavioral results. (**A**) Overview of the experimental design with the four conditions genuine vs. pretended, pain vs. no pain. Examples show static images, while in the experiment participants were shown video clips. (**B**) Overview of experimental timeline. At the outset of each block, a reminder of 'genuine' or 'pretended' was shown (both terms are shown here for illustrative purposes, in the experiment either genuine or pretended was displayed). After a fixation cross, a video in the corresponding condition appeared on the screen. Followed by a short jitter, three questions about the video were separately presented and had to be rated on a visual analogue scale. These would then be followed by the next video clip and questions (not shown). (**C**) Violin plots of the three types of ratings for all conditions. Participants generally demonstrated higher ratings for painful expressions in others, painful feelings in others, and unpleasantness in self in the genuine pain condition than in the pretended pain condition. Ratings of all three questions were higher in the painful situation than in the neutral situation, regardless of whether in the genuine or pretended condition. The thick black lines illustrate mean values, and the white boxes indicate a 95% CI. The dots are individual data, and the "violin" outlines illustrate their estimated density at different points of the scale. (**D**) Correlations of painful feelings in others and unpleasantness in self for the genuine pain and the pretended pain (the relevant questions were highlighted with a green rectangular). Results revealed a significant Pearson correlation between the two questions in the genuine pain condition, but no correlation in the pretended pain condition. The lines represent the fitted regression lines, bands indicate a 95% CI.

patterns of affective responses and self-other distinction for the genuine and pretended pain situations, focusing on the aIns, aMCC, and their interaction with rSMG. Furthermore, we investigated the relationship between neural activity and behavioral responses as well as empathic traits. In line with the literature reviewed above, we expected that, on the behavioral level, genuine pain would result in – alongside the obvious other-oriented higher pain ratings – higher self-oriented unpleasantness ratings. On the neural level, we predicted aIns and aMCC to show a stronger response to the genuine expressions of pain, but that these areas would also respond to the pretended pain, but to a lower extent. Differences in rSMG engagement and distinct patterns of this area's effective connectivity with aIns and aMCC were expected to relate to self-other distinction, and thus to explain the different empathic responses to genuine pain vs. pretended pain.

## Results

### Behavioral results

Three repeated-measures ANOVAs were performed with the factors *genuineness* (genuine vs. pretended and *pain* [pain vs. no pain]), for each of the three behavioral ratings. For ratings of painful *expressions* in others (***Figure 1C***, left), there was a main effect of the factor genuineness: participants showed higher ratings for the genuine vs. pretended conditions, $F_{genuineness}$ (1, 42) = 8.816, p=0.005, $\eta^2$ = 0.173. There was also a main effect of pain: participants showed higher ratings for the pain vs.

no pain conditions, $F_{pain}$ (1, 42) = 1718.645, p<0.001, $\eta^2$ = 0.976. The interaction term was significant as well, $F_{interaction}$ (1, 42) = 7.443, p=0.009, $\eta^2$ = 0.151, and this was related to higher ratings of painful expressions in others for the genuine pain compared to the pretended pain condition. For ratings of painful feelings in others (*Figure 1C*, middle), there was a main effect of genuineness: participants showed higher ratings for the genuine vs. pretended conditions, $F_{genuineness}$ (1, 42) = 770.140, p<0.001, $\eta^2$ = 0.948. There was also a main effect of pain, as participants showed higher ratings for the pain vs. no pain conditions, $F_{pain}$ (1, 42) = 1544.762, p<0.001, $\eta^2$ = 0.974. The interaction for painful feelings ratings was significant as well, $F_{interaction}$ (1, 42) = 752.618, p<0.001, $\eta^2$ = 0.947, and this was related to higher ratings of painful feelings in others for the genuine pain compared to the pretended pain condition. For ratings of unpleasantness in self (*Figure 1C*, right), there was a main effect of genuineness: participants showed higher ratings for the genuine vs. pretended conditions, $F_{genuineness}$ (1, 42) = 74.989, p<0.001, $\eta^2$ = 0.641. There was also a main effect of pain: participants showed higher ratings for the pain vs. no pain conditions, $F_{pain}$ (1,42) = 254.709, p<0.001, $\eta^2$ = 0.858. The interaction for unpleasantness ratings was significant as well, $F_{interaction}$ (1, 42) = 73.620, p<0.001, $\eta^2$ = 0.637, and this was related to higher ratings of unpleasantness in self for the genuine pain compared to the pretended pain condition. In sum, the behavioral data indicated higher ratings and large effect sizes of painful feelings in others and unpleasantness in self for the genuine pain compared to the pretended pain condition. Ratings of pain expressions also differed in terms of genuineness, at comparably low effect size, though they were expected to not show a difference by way of our experimental design and the pilot study.

We also found a significant correlation between behavioral ratings of painful feelings in others and unpleasantness in self in the genuine pain condition, r = 0.691, p<0.001, while in the pretended pain condition, the correlation was not significant, r = 0.249, p=0.107 (*Figure 1D*). A bootstrapping comparison showed a significant difference between the two correlation coefficients, p=0.002, 95 % Confidence Interval (CI) = [0.230, 1.060].

## fMRI results: mass-univariate analyses

Three contrasts were computed: (1) genuine: pain – no pain, (2) pretended: pain – no pain, and (3) genuine (pain – no pain) – pretended (pain – no pain). Across all three contrasts, we found activations as hypothesized in bilateral aIns, aMCC, and rSMG (*Figure 2A* and *Table 1*).

To identify whether or which brain activity was selectively related to the behavioral ratings described above, we performed a multiple regression analysis where we explored the relationship of activation

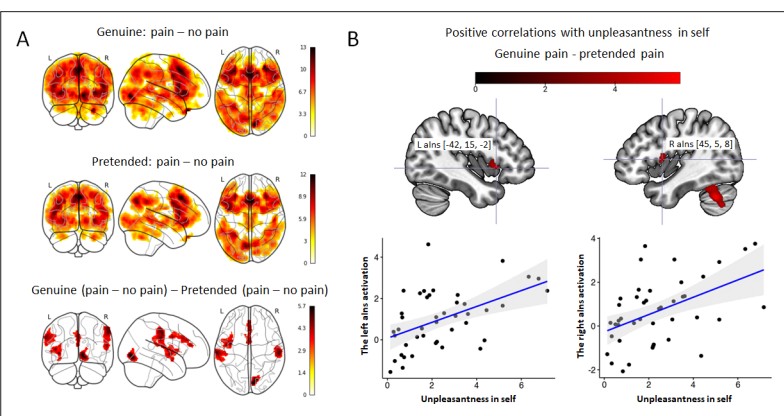

**Figure 2.** Neuroimaging results: mass-univariate analyses. (**A**) Activation maps of genuine: pain – no pain (top), pretended: pain - no pain (middle), and genuine (pain – no pain) – pretended (pain – no pain) (bottom). As expected, we found brain activations in the bilateral aIns, aMCC, and rSMG in all three contrasts (except for the bottom contrast, where the right aIns is only close to the significance threshold). (**B**) The multiple regression analysis demonstrated significant clusters in the left (peak: [–42, 15,–2]) and right anterior insular cortex (peak: [45, 5, 8]) that were positively correlated with the ratings of unpleasantness in self comparing genuine pain vs. pretended pain. All activations are thresholded with cluster-level family-wise error correction, p<0.05 (p<0.001 uncorrected initial selection threshold). The lines of the scatterplots represent the fitted regression lines, bands indicate a 95% confidence interval (CI).

**Table 1.** Results of mass-univariate functional segregation analyses in the MNI space.
Region names were labeled with the AAL atlas, threshold p<0.05 cluster-wise family-wise error correction (initial selection threshold p<0.001, uncorrected). BA = Brodmann area, L = left hemisphere, R = right hemisphere.

| Region label | BA | Cluster size | X | Y | Z | t-value |
|---|---|---|---|---|---|---|
| Genuine: pain - no pain | | | | | | |
| Lingual_R | 18 | 183,732 | 11 | −84 | −3 | 13.38 |
| Temporal_Pole_Sup_R | 38 | | 30 | 33 | −33 | 13.31 |
| Supp_Motor_Area_R | 8 | | 5 | 15 | 51 | 12.96 |
| Supp_Motor_Area_R | 8 | | 3 | 17 | 50 | 12.92 |
| Supp_Motor_Area_L | 8 | | −5 | 17 | 48 | 12.56 |
| Insula_L | 45 | | −32 | 26 | 6 | 12.32 |
| Insula_R | 45 | | 33 | 29 | 3 | 12.09 |
| Frontal_Inf_Oper_R | 44 | | 51 | 14 | 15 | 12.01 |
| Frontal_Inf_Oper_R | 44 | | 50 | 12 | 18 | 11.79 |
| Precentral_L | 6 | | −42 | 3 | 39 | 11.72 |
| Fusiform_R | 20 | 463 | 36 | −5 | −41 | 5.58 |
| Pretended: pain - no pain | | | | | | |
| Supp_Motor_Area_R | 8 | 59,665 | 5 | 20 | 48 | 11.80 |
| Supp_Motor_Area_L | 8 | | −6 | 18 | 50 | 11.14 |
| Frontal_Inf_Oper_L | 44 | | −50 | 15 | 15 | 10.39 |
| Insula_R | 45 | | 33 | 29 | 0 | 9.81 |
| Insula_L | 45 | | −29 | 30 | 0 | 9.60 |
| Frontal_Inf_Tri_R | 44 | | 47 | 15 | 26 | 9.21 |
| Precuneus_L | 7 | 35,136 | −9 | −71 | 41 | 10.27 |
| Parietal_Inf_L | 39 | | −32 | −51 | 41 | 9.39 |
| Precuneus_R | 7 | | 9 | −69 | 38 | 8.44 |
| Temporal_Mid_L | 21 | | −53 | −47 | 5 | 7.67 |
| Occipital_Mid_L | 19 | | −44 | −78 | 2 | 7.47 |
| Parietal_Inf_R | 39 | | 39 | −50 | 41 | 7.25 |
| Temporal_Mid_R | 22 | 12,970 | 51 | −20 | −6 | 7.70 |
| Lingual_R | 17 | | 12 | −86 | −2 | 7.40 |
| Fusiform_R | 37 | | 47 | −33 | −27 | 5.32 |
| Occipital_Mid_R | 18 | | 33 | −86 | 3 | 5.23 |
| Cingulum_Mid_R | 23 | 1666 | −3 | −14 | 27 | 6.35 |
| Cingulum_Mid_L | 23 | | −3 | −24 | 32 | 5.57 |
| Temporal_Pole_Sup_R | 47 | 589 | 32 | 35 | −33 | 7.18 |
| Frontal_Sup_Orb_R | 11 | | 17 | 41 | −24 | 3.36 |
| Genuine (pain – no pain) – pretended (pain – no pain) | | | | | | |
| SupraMarginal_L | 40 | 1877 | −66 | −21 | 32 | 4.94 |
| Postcentral_L | 1 | | −50 | −21 | 26 | 3.75 |

*Table 1 continued on next page*

*Table 1 continued*

| Region label | BA | Cluster size | X | Y | Z | t-value |
|---|---|---|---|---|---|---|
| SupraMarginal_R | 40 | 1833 | 63 | −20 | 42 | 5.09 |
| Rolandic_Oper_R | 40 | | 59 | −15 | 14 | 4.47 |
| Insula_L | 13 | 1299 | −38 | −3 | −2 | 5.01 |
| Rolandic_Oper_L | 4 | | −45 | −6 | 8 | 4.8 |
| Cingulum_Ant_L | 32 | 1138 | 0 | 41 | 17 | 4.54 |
| Cingulum_Mid_R | 32 | | 2 | 24 | 32 | 4.45 |
| Cingulum_Mid_L | 24 | | 0 | 2 | 35 | 4.43 |
| Cingulum_Ant_R | 8 | | 2 | 32 | 27 | 4.42 |
| Lingual_R | 18 | 1003 | 9 | −84 | −3 | 5.72 |
| Calcarine_R | 17 | | 18 | −78 | 8 | 3.61 |
| Insula_R | 13 | 225 | 39 | 8 | −3 | 3.91 |
| Rolandic_Oper_R | 13 | | 41 | 0 | 11 | 3.77 |

in the contrast genuine pain – pretended pain with the three behavioral ratings. We found significant clusters in bilateral aIns, visual cortex, and cerebellum that could be selectively explained by the ratings of self-unpleasantness rather than ratings of painful expressions in others or painful feelings in others (*Figure 2B*).

## DCM results

We performed DCM analysis to specifically examine the modulatory effect of genuineness on the effective connectivity between the right aIns and rSMG. More specifically, we sought to assess whether the experimental manipulation of genuine pain vs. pretended pain tuned the bidirectional neural dynamics from aIns to rSMG and vice versa, in terms of both directionality (sign of the DCM parameter) and intensity (magnitude of the DCM parameter). If the experimental manipulation modulated the effective connectivity, we would observe a strong posterior probability ($p_p$ >0.95) of the modulatory effect. Our original analysis plan was to include aMCC in the DCM analyses, but based on the fact that aMCC did not show as strong evidence (in terms of the multiple regression analysis) as the aIns of being involved in our task, we decided to use a more parsimonious DCM model without the aMCC.

We found strong evidence of inhibitory effects on the aIns to rSMG connection both in the genuine pain condition and in the pretended pain condition (*Figure 1A-C*). Comparing the strength of these modulatory effects on the aIns to rSMG connection revealed a reduced inhibitory effect for genuine pain as opposed to pretended pain, $t_{41}$ = 2.671, p=0.011 (mean$_{genuine\ pain}$ = −0.821, 95% CI = [−0.878, −0.712]; mean$_{pretended\ pain}$ = −0.934, 95% CI = [−1.076, −0.822]; *Figure 3C*). There was no evidence of a modulatory effect on the rSMG to aIns connection.

## Individual associations between modulatory effects, behavioral ratings, and questionnaires

To examine how the modulatory effects from the DCM were related to the behavioral ratings, we computed two multiple linear regression models for each condition. For the genuine pain condition, we find that the modulatory effect was significantly related to the rating of painful feelings in others (t = 2.317, p=0.026), but not related to the rating of either painful expressions in others (t = −1.492, p=0.144) or unpleasantness in self (t = 0.058, p=0.954). For the pretended pain condition, none of the ratings was significantly related to the modulatory effect (*Figure 3D*). The variance inflation factors (VIFs) for three ratings in both models were calculated to diagnose collinearity, showing no severe collinearity problem (all VIFs < 5; the smallest VIF = 1.132 and the largest VIF = 4.387).

In addition, we tested two multiple stepwise linear regression models to investigate whether subscales of all three questionnaires could explain modulatory effects for genuine pain and pretended pain. In the genuine pain condition, we found that the modulatory effect was significantly explained

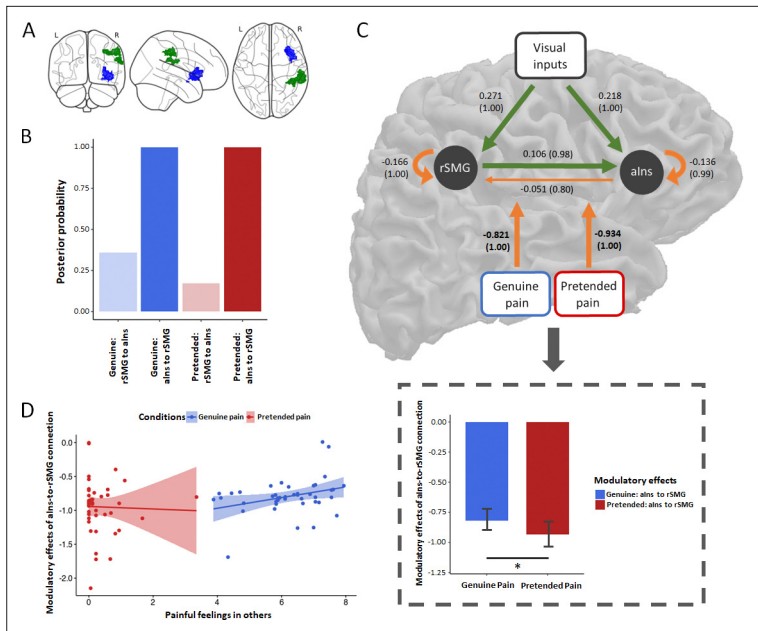

**Figure 3.** DCM results and brain-behavior analyses. (**A**) ROIs included in the DCM: aIns (blue; peak: [33, 29, 2]) and rSMG (green; peak: [41, −39, 42]). (**B**) Posterior probability of modulatory effects for the genuine pain and the pretended pain. (**C**) The group-average DCM model. Green arrows indicate neural excitation, and orange arrows indicate neural inhibition. Importantly, we found strong evidence of inhibitory effects on the connection of aIns to rSMG for both the genuine pain condition and the pretended pain condition. Values without the bracket quantify the strength of connections and values in the bracket indicate the posterior probability of connections. All DCM parameters of the optimal model showed greater than a 95 % posterior probability (i.e., strong evidence) except for the intrinsic connection of aIns to rSMG ($p_p$ = 0.80). Paired sample t-test showed less inhibitory effects of the aIns-to-rSMG connection for the genuine pain than the pretended pain. This result is highlighted with a gray rectangular. Data are mean ± 95% CI. (**D**) The multiple linear regression model revealed a positive correlation between the inhibitory effect and painful feelings in others and not with the other two ratings for genuine pain but no correlation for pretended pain.

by scores of two subscales, that is affective ability and affective reactivity of the ECQ: $F_{model}$ (1, 39) = 6.829, p=0.003, $R^2$ = 0.270; $B_{affective\ ability}$ = 0.052, beta = 0.497, p=0.002; $B_{affective\ reactivity}$ = −0.040, beta = −0.421, p=0.008. No significant predictor was found with the other questionnaires (i.e., IRI and TAS). In the pretended pain condition, none of the three questionnaires significantly predicted variations of the modulatory effect. No severe collinearity problem was detected for either regression model (all VIFs < 2; the smallest VIF = 1.011 and the largest VIF = 1.600).

## Discussion

In this study, we developed and used a novel experimental paradigm in which participants watched video clips of persons who supposedly either genuinely experienced pain or merely pretended to be in strong pain. Combining mass-univariate analysis with effective connectivity (DCM) analyses, our study provides evidence on the distinct neural dynamics between regions suggestive of affect processing (i.e., aIns and aMCC) and self-other distinction (i.e., rSMG) for genuinely sharing vs. responding to pretended, non-genuine pain. With this, we aimed to clarify two main questions: First, whether neural responses in areas such as the aIns and aMCC to the pain of others are indeed related to a veridical sharing of affect, as opposed to simply tracking automatic responses to salient affective displays. And second, how processes related to self-other distinction, implemented in the rSMG, enable appropriate empathic responses to genuine vs. merely pretended affective states.

The mass-univariate analyses suggest that the increased activity in aIns for genuine pain as opposed to pretended pain properly reflects affect sharing. As aforementioned, the network of affective sharing and certain domain-general processes (e.g., salience detection and automatic emotion processing) overlap in aIns and aMCC (*Zaki et al., 2016*, for review). This indicates that indeed, part

of the activation in these areas could be related to perceptual salience, which is why it has been widely debated as a potential confound of empathy and affect sharing models (*Zaki et al., 2016*, for review; *Lamm et al., 2019*, for review). However, when comparing genuine pain versus pretended pain, activity in these areas was not only found to be stronger in response to genuine pain, but the increased activation in aIns was also selectively correlated with ratings of self-oriented unpleasantness and was not correlated with either other-related painful expressions or painful feelings in terms of the regression analysis. That only aIns and not also aMCC shows such correlation may be explained by previous studies, according to which aIns is more specifically associated with affective representations, while the role of aMCC rather seems to evaluate and regulate emotions that arise due to empathy (*Fan et al., 2011*; *Lamm et al., 2011*; *Jauniaux et al., 2019*). Taken together, the activation and brain-behavior findings provide evidence that responses in aIns (and to a lesser extent also the aMCC) are not simply automatic responses triggered by perceptually salient events (otherwise the increased aIns activation should also be explained by other behavioral ratings in the sense of shared influence by domain-general effects). Rather, they seem to track the actual affective states of the other person, and thus the shared neural representation of that response (see *Zhou et al., 2020*, for similar recent conclusions based on multi-voxel pattern analyses). Our findings are also in line with the proposed model of *Coll et al., 2017*, which suggests that affect sharing is the consequence of emotion identification. More specifically, while part of the activation in the aIns and aMCC is indeed related to an (presumably earlier) automatic response, the added engagement of these areas once they have identified the pain as genuine shows that only in this condition, they then also engage in proper affect sharing. Ideally, one should be able to discern these processes in time, but neither the temporal resolution of our fMRI measurements nor the paradigm in which we always announced the conditions beforehand would have been sensitive enough to do so. Thus, future studies including complementary methods, such as EEG and MEG, and tailored experimental designs are needed to pinpoint the exact sequence of processes engaged in automatic affective responses vs. proper affect sharing.

Beyond higher activation in affective nodes supporting (pain) empathy, increased activation was also found in rSMG. The inferior parietal lobule was shown to be generally engaged in selective attention, action observation, and imitating emotions (*Bach et al., 2010*; *Pokorny et al., 2015*; *Gola, 2017*; *Hawco et al., 2017*). Importantly, a specific role in affective rather than cognitive self-other distinction has been identified for rSMG (*Silani et al., 2013*; *Steinbeis et al., 2015*; *Bukowski et al., 2020*). Based on such findings, it has been proposed that the rSMG allows for a rapid switching between or the integration of self- and other-related representations, as two processes that may underpin the functional basis of successful self-other distinction (*Lamm et al., 2016*, for review). Theoretical models of empathy and related socio-affective responses suggest that such regulation is especially important to avoid so-called empathic over-arousal, which would shift the focus away from empathy and the other's needs, toward taking care of one's own personal distress (*Batson et al., 1987*, for review; *Decety and Lamm, 2011*, for review). Concerning the current findings, we thus propose that the higher rSMG engagement in the genuine pain condition reflects an increasing demand for self-other distinction imposed by the stronger shared negative affect experienced in this condition.

Beyond these differences in the magnitude of rSMG activation, the DCM analysis demonstrated less inhibition on the aIns-to-rSMG connection for genuine pain compared to pretended pain. Note that our focus on the right aIns rather than bilateral aIns was because it is located in the same hemisphere as the right SMG. Various theoretical accounts suggest that areas such as the aIns and rSMG may play a key role in comparing self-related information with the sensory evidence (*Decety and Lamm, 2007*, for review; *Seth, 2013*, for review). According to recent theories on predictive processing (*Clark, 2013*, for review) and active inference (*Friston, 2010*, for review), the brain can be regarded as a "prediction machine", in which the top-down signals pass over predictions and the bottom-up signals convey prediction errors across different levels of cortical hierarchies (*Chen et al., 2009*; *Friston, 2010*, for review; *Bastos et al., 2015*). It is suggested that these top-down predictions are mediated by inhibitory neural connections (*Zhang et al., 2008*; *Bastos et al., 2015*; *Miska et al., 2018*). Our findings align with such views, by suggesting that the inhibitory connection from aIns to rSMG can be explained as the predictive mismatch between the top-down predictions of self-related information (e.g., personal affect) and sensory inputs (e.g., pain facial expressions). This suppression of neural activity leads to an *explaining away* of incoming bottom-up prediction error. This is reflected by the absence of any condition-dependent modulatory effects on the rSMG to aIns connection,

suggesting that the influence of the task conditions is sufficiently modeled by the predictions from aIns to rSMG. Therefore, the stronger inhibition for pretended pain, compared to genuine pain, could indicate a higher demand to overcome the mismatch between the visual inputs and the agent's prior beliefs and contextual information about the situation (i.e., "this person looks like in pain, but I know he/she does not actually feel it"). We speculate that a dynamic interaction between sensory-driven and control processes is underlying the modulatory effect: when individuals realized after an initial sensory-driven response to the facial expression that it was not genuinely expressing pain, control, and appraisal processes led to a reappraisal of the triggered emotional response, and thus a dampening of the unpleasantness. The reduced inhibition in the genuine pain condition could moreover be a mechanism that explains the higher rSMG activation in this condition.

Model comparison showed that the best model to explain the inhibitory effect with the behavioral ratings for both the genuine and pretended pain is the model without interactions between ratings. That is, if any behavioral rating contributed to the modulation of aIns to rSMG, the effect would be more likely coming from single ratings rather than their interactions. Specifically, we found the strength of the inhibitory effect in the genuine pain condition to correlate with ratings of painful feelings in others, but not with the ratings of pain expression in others or unpleasantness in self. For the pretended pain condition, none of the ratings showed a correlation. The latter could in principle be due to a lack of variation in the ratings (which by way of the design were mostly close to zero or one). We deem it more plausible, though, that the correlation findings provide further evidence that the modulation of aIns to rSMG is implicated in encoding others' emotional states, which serves as a functional foundation for self-other processing when participants engaged in genuine affect sharing. This regulation cannot be totally attributed to domain-general processes, otherwise other ratings should have also explained this variation. It is also interesting to note that the found correlation relates to cognitive evaluations of the other's pain rather than to own affect, as tracked by the unpleasantness in self-ratings. This would to some extent be in line with DCM findings by *Kanske et al., 2016*. These authors found that the inhibition of the temporoparietal junction (TPJ) by the aIns was linked to interactions between Theory of Mind (ToM) and empathic distress, that is the interaction of 'cognitive' vs. 'affective' processes engaged in understanding others' cognitive and affective states. Note that the right TPJ is an overarching area involved in self-other distinction of which rSMG is considered a part or at least closely connected to *Decety and Lamm, 2007*, for review.

The correlations between the DCM inhibitory effect and empathic traits assessed via questionnaires provide further refinements for the relevance of rSMG in implementing self-other distinction to allow for an appropriate empathic response. When participants shared genuine affect, the inhibitory effect on the aIns to rSMG connection was positively correlated with affective ability and negatively correlated with affective reactivity. Affective ability reflects the capacity to subjectively share emotions with others, while affective reactivity plays a role in the susceptibility to vicarious distress and thus to more automatic responses to another's emotion (*Batchelder et al., 2017*). Again, as for the correlations with the three rating scales, we did not find correlations of empathic traits for the pretended pain condition. Taken together, the DCM results and their qualification by the correlation findings suggest that in the genuine pain condition, which requires an accurate sharing of pain, rSMG interacts with aIns to achieve 'affective-to-affective' self-other distinction – that is disambiguating affective signals originating in the self from those attributable to the other person. The aIns to rSMG connection in the pretended pain condition may reflect a related, yet slightly distinct mechanism. Here, it seems that 'cognitive-to-affective' self-other distinction is at play, which helps resolve conflicting information between the top-down contextual information (i.e., that the demonstrator is not actually in pain) from what seems an unavoidable affective response to the highly salient perceptual cue of the facial expression of pain. Given our behavioral and trait data did not allow us to distinguish more precisely between these different types of self-other distinction, this however remains an interpretation and a hypothesis that will require further investigation. This inhibitory effect might be related to socioemotional disturbances of individuals with autism spectrum disorders (ASD), who show impairments in social cognition, including self- and other-related processing (*Hoffmann et al., 2016*; *Lamm et al., 2016*, for review). It is thus likely that ASD individuals exhibit distinct inhibition of the aIns-to-rSMG connectivity pattern compared to healthy controls. Further research with ASD individuals is required.

One potential limitation of the study could be the slightly higher ratings of other-oriented pain expressions for genuine pain, which were hypothesized to have no difference, as compared to

pretended pain. As we found the enhanced aIns activation in the genuine pain condition mainly tracked personal unpleasantness rather than perceptually domain-general processes, and because the effect size of the pain expression difference was much smaller than for the affect ratings, we consider this difference did not fundamentally influence the interpretation of our findings. An additional limitation was that our study design did not aim to explicitly quantify self-other distinction. Rather, in line with previous research and based on our theoretical framework and rationale, we inferred the engagement of this process from the experimental conditions and the associated behavioral and neural responses. We expect our findings to prompt and inform future research designed to quantify and experimentally disentangle self- and other-related processes more explicitly. Note though that our participants and the targets shown in the videos were balanced with respect to their sex/gender. Yet sex/gender effects (for a review, though, see *Christov-Moore, 2014*) were outside the scope of the current study, and we thus did not perform any sex/gender-related analyses.

In conclusion, the current study advances our understanding of two main aspects of empathy. First, we provide evidence that empathy-related responses in the aIns can indeed be linked to affective sharing, rather than attributing them to responses triggered only by perceptual saliency. Second, we show how aIns and rSMG are orchestrated to track what another person really feels, thus enabling us to appropriately respond to their actual needs. Beyond these basic research insights, our study provides novel avenues for clinical application, and the investigation of contextual and interpersonal factors in the accurate diagnosis of pain and its expression.

## Materials and methods

### Participants

Forty-eight participants took part in the study. Five of them were excluded because of excessive head motion ( >15 % scans with the frame-wise displacement over 0.5 mm in one session). Data of the remaining 43 participants (21 females; age: Mean = 26.72 years, S.D. = 4.47) were entered into analyses. This sample size was determined on a priori power analysis in Gpower 3.1 (*Faul et al., 2007*). We assumed a medium effect size of Cohen's $d$ = 0.5. After calculation, the minimum sample size statistically required for this study was 34 ($\alpha$ = 0.05, two-tailed, $1-\beta$ = 0.80). Participants were pre-screened by an MRI safety-check questionnaire, assuring normal or corrected to normal vision and no presence or history of neurologic, psychiatric, or major medical disorders. All participants were being right-handed (self-reported) and provided written consent including post-disclosure of any potential deception. The study was approved by the ethics committee of the Medical University of Vienna and was conducted in line with the latest version of the *World Medical Association, 2013*.

### Manipulation of facial expressions

As part of our study, we developed a novel experimental design and corresponding stimuli, which consisted of video clips showing different demonstrators ostensibly in four different situations: (1) Genuine pain: the demonstrator's right cheek was penetrated by a hypodermic needle attached to a syringe, and the demonstrator's facial expression changed from neutral to a strongly painful facial expression. (2) Genuine no pain: the demonstrator maintained a neutral facial expression when a Q-tip fixed on the backend of the same syringe touched their right cheek. (3) Pretended pain: the demonstrator's right cheek was approached by the same syringe and the hypodermic needle, with the latter covered by a protective cap; upon touch by the cap, the demonstrator's facial expression changed from neutral to a strongly painful facial expression. (4) Pretended no pain: the demonstrator maintained a neutral facial expression when a Q-tip fixed on the backend of the same syringe touched their right cheek.

To create these stimuli, we recruited 20 demonstrators (10 females), with experience in acting, and filmed them in front of a dark blue background. An experimenter who stood on the right side of the demonstrators, but of whom only the right hand holding the syringe could be seen, administered the injections and touches. Unbeknownst to the participants, all painful expressions were acted, as the needle was a telescopic needle (i.e., a needle that seemed to enter the cheek upon contact, but in reality, was invisibly retracting into the syringe). The reason for using a protective cap in the pretended pain condition was to match the perceptual situation that an aversive object was approaching a body part in both pain conditions. In all situations, the demonstrator was instructed to look naturally toward

the camera 1.5 m in front of them. As soon as the needle or the cap touched the demonstrator's cheek, the demonstrator made a painful facial expression, as naturally and vividly as possible. In the neutral control conditions, demonstrators maintained a neutral facial expression when a Q-tip fixed at the backend of the syringe touched their cheek. Again, a syringe with a needle attached to the other end was used to perceptually control for the presence of an aversive object in all four conditions. Note that in another set of conditions, demonstrators showed disgusted or neutral expressions. Data from these conditions will be reported elsewhere. All demonstrators signed an agreement that their video clips and static images could be used for scientific purposes.

## Stimulus validation and pilot study

To validate the stimuli, we performed an online validation study with N = 110 participants, who were asked to rate a total of 120 video clips of 2 s duration of the two conditions (60 of each condition) showing painful expressions (i.e., the genuine and the pretended pain conditions). The main aim of the validation study was to identify a set of demonstrators that expressed pain with comparable intensity and quality, and whose pain expressions in the genuine and pretended conditions were comparable. After each video clip, participants rated three questions on a visual analog scale with nine tick-marks and the two end-points marked as 'almost not at all' to 'unbearable': (1) How much pain did the person *express* on his/her face? (2) How much pain did the person *actually* feel? (3) How *unpleasant* did you feel to watch the person in this situation? The order of these three questions was pseudo-randomized. Moreover, eight catch trials randomly interspersed across the validation study to test whether participants maintained attention to the stimuli. Here, participants were asked to correctly select the demonstrator they had seen in the last video, between two static images of the correct and a distractor demonstrator displayed side by side, both showing neutral facial expressions.

The validation study was implemented within the online survey platform SoSci Survey (https://www.soscisurvey.de), with a study participation invite published on Amazon Mechanical Turk (https://www.mturk.com/), a globally commercial platform allowing for online testing. Survey data of 62 of 110 participants (34 females; age: mean = 28.71 years, S.D. = 10.11) were entered into analysis (inclusion criteria: false rate for the test questions < 2/8, survey duration > 20 min and <150 min, and the maximum number of continuous identical ratings < 5). Based on this validation step, we had to exclude videos of six demonstrators ( three females) for which participants showed a significant difference in painful expressions in others between the genuine pain and the pretended pain conditions. As a result of this validation, videos of 14 demonstrators (7 females), which showed no difference in the pain *expression* rating between genuine and pretended conditions and which overall showed comparable mean ratings in all three ratings, were selected for the subsequent pilot study.

In the pilot study, 47 participants (24 females; age: mean = 26.28 years, S.D. = 8.80) were recruited for a behavioral experiment in the behavioral laboratory. The aim was to verify the experimental effects and the feasibility of the experimental procedures that we intended to use in the main fMRI experiment, as well as to identify video stimuli that may not yield the predicted responses. Thus, all four conditions described above were presented to the participants. Participants were explicitly instructed that they would watch other persons' genuine painful expressions in some blocks, while in other blocks, they would see other persons acting out painful expressions (recall that in reality, all demonstrators had been actors, and the information about this type of necessary deception was conveyed to participants at the debriefing stage). They would see all demonstrators' neutral expressions as well. Participants were instructed to rate the three questions mentioned above. Upon screening for video clips that showed aberrant responses, we excluded videos of two demonstrators (one female), for whom the pain *expression* rating difference between the pretended vs. genuine expressions was large. Forty-eight videos of 12 demonstrators entered the following analyses. Three separate repeated-measures ANOVAs were respectively performed for the three rating questions. For the main effect of *genuineness* (genuine vs. pretended), it was not significant and low in effect size for painful expressions in others ($F_{genuineness}$ (1, 46) = 2.939, p=0.093, $\eta^2$ = 0.060), but was significant with high effect size for the painful feelings in others ($F_{genuineness}$ (1, 46) = 280.112, p<0.001, $\eta^2$ = 0.859) as well as the unpleasantness in self ($F_{genuineness}$ (1, 46) = 43.143, p<0.001, $\eta^2$ = 0.484). The main effects of *pain* (pain vs. no pain) for all three questions were found significant with high effect size (the smallest effect size was for the rating of unpleasantness in self, $F_{pain}$ (1, 46) = 82.199, p<0.001, $\eta^2$ = 0.641). Our pilot study thus (1) provided assuring evidence that the novel experimental paradigm

worked as expected and (b) made it possible to select video clips that we could match for the two conditions (i.e., genuine pain and pretended pain). More specifically, as expected and required for the main study, participants rated the painfulness of the demonstrators to be substantially higher when it was genuine as compared to those that were pretended, and this also resulted in much higher unpleasantness experienced in the self. It is worth noting that, the two conditions did not differ with respect to the ratings of the painful facial expressions, implying that putative differences in ratings as well as the subsequent brain imaging data could only be attributed to the contextual appraisal of the demonstrators' actual painful states, rather than the differences in facial pain perception. Based on this pilot study, we thus decided on video clips of 12 demonstrators (6 females) in the main fMRI experiment.

## Experimental design and procedure of the fMRI study

The experiment was implemented using Cogent 2000 (version 1.33; http://www.vislab.ucl.ac.uk/cogent_2000.php). MRI scanning took place at the University of Vienna MRI Center. Once participants arrived at the scanner site, an experimenter instructed them that they would watch videos from the four conditions outlined above. Participants were explicitly instructed to recreate the feelings of the demonstrators shown in the videos as vividly and intensely as possible. Based on the validation and pilot study, the painful *expressions* for the genuine and pretended conditions were matched. We also counterbalanced the demonstrators appearing in the genuine and pretended conditions across participants, thus controlling for differences in behavioral and brain response that could be explained by differences between the stimulus sets. Note that, all video clips were validated and piloted multiple times to ensure the experimental effect (details can be found in the section above).

The participant performed the fMRI experiment in two runs (*Figure 1*). Each run was composed of two blocks showing genuine pain and two blocks showing pretended pain. In each block, the participant watched nine video clips containing both painful and neutral videos. To remind participants' the condition of the upcoming block, a label of 4 s duration appeared at the beginning of each block, showing either 'genuine' or 'pretended' (in German). Each trial started with a fixation cross (+) presented for 4–7 s (in steps of 1.5 s, mean = 5.5 s). After that, the video (duration = 2 s) was played. A short jitter was inserted after the video for 0.5–1.0 s (in steps of 0.05 s, mean = 0.75 s). After the jitter, the following three questions were displayed (in German) one after the other in a pseudo-randomized order: (1) How much pain did the person *express* on his/her face? (2) How much pain did the person *actually feel*? (3) How *unpleasant did you feel* to watch the person in this situation? Beneath each question, a visual analog scale ranging from 0 (not at all) to 8 (unbearable) with nine tick-marks was positioned. The participant moved the marker along the scale by pressing the left or right keys on the button box, and they pressed the middle key to confirm their answer. The marker initially was always located at the midpoint ('4') of the scale. When the confirmed key was pressed, the marker turned from black to red. All ratings lasted for 4 s even when the participant pressed the confirmed key before the end of this period. Between the two runs, the participant had a short break (1–2 min).

Before entering the scanner, participants conducted practice trials on the computer to get familiarized with the button box and the experimental interface. After that, participants were moved into the scanner and performed the task. Following the functional imaging runs, a 6.5 min structural scanning was employed. When participants finished the scanning session, they were scheduled for a date to complete three questionnaires in the lab: the Empathy Components Questionnaire (ECQ) (*Batchelder, 2015*; *Batchelder et al., 2017*), the Interpersonal Reactivity Index (IRI) (*Davis, 1980*), and the Toronto Alexithymia Scale (TAS) (*Bagby et al., 1994*). For the ECQ, there are 27 items in total to be categorized into five subscales: cognitive ability, cognitive drive, affective ability, affective drive, and affective reactivity, using a 4-point Likert scale ranging from 1 ('strongly disagree') to 4 ('strongly agree') (*Batchelder, 2015*; *Batchelder et al., 2017*). For the IRI, there are 28 items divided into four subscales: perspective taking, fantasy, empathic concern, and personal distress, using a 5-point Likert scale ranging from 0 ('does not describe me well') to 4 ('describes me very well') (*Davis, 1980*). For the TAS, there are 20 items and three subscales – difficulty describing feelings, difficulty identifying feelings, and externally oriented thinking, using a 5-point Likert scale ranging from 1 ('strongly disagree') to 5 ('strongly agree') (*Bagby et al., 1994*). The average interval between the scanning session and the lab survey was 1 week. The participant was debriefed after completing the whole study.

## Behavioral data analysis

We applied repeated-measures ANOVAs to investigate the main effects and the interaction of the two factors genuine vs. pretended and pain vs. no pain, using SPSS (version 26.0; IBM). Furthermore, we conducted Pearson correlations to examine whether ratings of painful feelings in others were correlated with unpleasantness in self for the genuine pain and the pretended pain. The correlation coefficients were further compared using a bootstrap approach with the R package bootcorci [https://github.com/GRousselet/bootcorci (*Rousselet, 2021*)].

## fMRI data acquisition

fMRI data were collected using a Siemens Magnetom Skyra MRI scanner (Siemens, Erlangen, Germany), with a 32-channel head coil. Functional whole-brain scans were collected using a multiband-accelerated T2*-weighted echoplanar imaging sequence (multiband acceleration factor = 4, interleaved ascending acquisition in multi-slice mode, 52 slices co-planar to the connecting line between anterior and posterior commissure, TR = 1200 ms, TE = 34 ms, acquisition matrix = 96 × 96 voxels, FOV = 210 × 210 mm$^2$, flip angle = 66°, inter-slice gap = 0.4 mm, voxel size = 2.2 × 2.2 × 2 mm$^3$). Two functional imaging runs, each lasting around 16 min (~800 images per run), were performed. Structural images were acquired with a magnetization-prepared rapid gradient-echo sequence (TE/TR = 2.43/2300 ms, flip angle = 8°, ascending acquisition, single-shot multi-slice mode, FOV = 240 × 240 mm$^2$, voxel size = 0.8 × 0.8 × 0.8 mm$^3$, 208 sagittal slices, slice thickness = 0.8 mm).

## fMRI data processing and mass-univariate functional segregation analyses

Imaging data were preprocessed with a combination of Nipype (*Gorgolewski et al., 2011*) and MATLAB (version R2018b 9.5.0; MathWorks), with Statistical Parametric Mapping (SPM12; https://www.fil.ion.ucl.ac.uk/spm/software/spm12/). Raw data were imported into BIDS format (http://bids.neuroimaging.io/). Functional data were subsequently preprocessed using slice timing correction to the middle slice (*Sladky, 2011*), realignment to the first image of each session, co-registration to the T1 image, segmentation between gray matter, white matter, and cerebrospinal fluid, normalization to MNI template space using Diffeomorphic Anatomical Registration Through Exponentiated Lie Algebra (DARTEL) toolbox (*Ashburner, 2007*), and smoothing with a 6 mm full width at half-maximum three-dimensional Gaussian kernel.

To improve data quality, we performed data scrubbing of the functional scans for those whose frame-wise displacements (FD) were over 0.5 mm (*Power et al., 2012*; *Power et al., 2014*). In other words, we identified individual outlier scans and flagged the volume indices as nuisance regressors in the general linear model (GLM) for the first-level analysis.

In order to perform mass-univariate functional segregation analyses, a first-level GLM design matrix was created and composed of two identically modeled runs for each participant. Seven regressors of interest were entered in each model: stimulation phase of the four conditions (i.e., genuine pain, genuine no pain, pretended pain, pretended no pain; 2000 ms) and rating phase of the three questions (i.e., painful expressions in others, painful feelings in others, and unpleasantness in self; 12,000 ms). Six head motion parameters and the scrubbing regressors (FD > 0.5 mm, if applicable) were additionally entered as nuisance regressors. Individual contrasts of the four conditions and the three ratings (all across the two runs) against implicit baseline were respectively created.

On the second level, a flexible factorial design was employed to perform the group-level analysis. The design included three factors: a between-subject factor (i.e., subject) that was specified independent and with equal variance, a within-subject factor (i.e., genuine or pretended) that was specified dependent and with equal variance, and a second within-subject factor (i.e., pain or no pain) that was specified dependent and with equal variance (*Gläscher and Gitelman, 2008*). Three contrasts were computed: (1) main effect of genuine: pain – no pain, (2) main effect of pretended: pain – no pain, and (3) interaction: genuine (pain – no pain) – pretended (pain – no pain). We applied an initial threshold of p<0.001 (uncorrected) at the voxel level and a family-wise error (FWE) correction (p<0.05) at the cluster level. The cluster extent threshold was determined by the SPM extension 'cp_cluster_Pthresh.m' (https://goo.gl/kjVydz).

## Brain-behavior relationships

A multiple regression model was built on the group level to investigate the relationship between specific brain activations and behavioral ratings. In this model, the contrast genuine pain – pretended pain was set as the dependent variable, and differences between conditions for three behavioral ratings were specified as independent variables. The reason that we used the comparison between conditions for both brain signals and behavioral ratings was to control for potential effects of perceptual salience. All covariates were mean-centered. An intercept was added in the model. To test whether the order of entering ratings into the regression model influence the results, we performed five additional regression analyses with all possible orders of three ratings. The results were consistent across all six regression models, and we only showed the result for one regression (i.e., expression+ feeling + unpleasantness) in the Results section. Note that, we performed the regression model with the contrast genuine pain – pretended pain instead of the more exhaustive contrast genuine (pain - no pain) - pretended (pain – no pain), and this was because the genuine and the pretended pain conditions were the main focus of our work. Moreover, the pain contrast showed more robust (in terms of statistical effect size) and widespread activations across the brain, making it more likely to pick up possible brain-behavior relationships. The same threshold as above was applied in this analysis.

We aimed to assess these brain-behavior relationships for the following regions of interest (ROI): (1) aIns and aMCC, that is two regions associated with affective processes and specifically with empathy for pain, (2) rSMG, an area implicated in affective self-other distinction. The ROI masks were defined as the conjunction of the averaging contrast between genuine and pretended: pain – no pain (threshold: voxel-wise FWE correction, $P < 0.05$) and the anatomical masks created by the Wake Forest University (WFU) Pick Atlas SPM toolbox (http://fmri.wfubmc.edu) with the automated anatomical atlas (AAL). The ROI masks were created with Marsbar ROI Toolbox implemented in SPM12 (*Brett et al., 2002*). Note that we specifically selected the ROIs this way, such that they were orthogonal (i.e., independent) to the subsequent analyses of interest. As exploratory analyses found significant correlations mainly in aIns, rather than in aMCC, we will focus in the results section on two ROIs: the right aIns and the rSMG. Focusing on the right aIns instead of the left one was because the right aIns is on the ipsilateral hemisphere as rSMG.

## Analyses using DCM

To investigate the functional network involved in affective processes and self-other distinction and how it was modulated by our experimental manipulations (i.e., genuine pain and pretended pain), we used DCM to estimate the effective connectivity between the ROIs based on the tasked-related brain responses (*Stephan and Friston, 2010*, for review). The DCM analyses were conducted with DCM12.5 implemented in SPM12 (v. 7771). First, we extracted individual time series separately for each ROI. To ensure the selected voxels engaged in a task-relevant activity but not random signal fluctuations, we determined the voxels both on a group-level threshold and an individual-level threshold (*Holmes et al., 2020*). An initial threshold was set as p<0.05, uncorrected. The significant voxels in the main effect of genuine pain and pretended pain were further selected by an individual threshold. For each participant, an individual peak coordinate within the ROI mask was searched, and an individual mask was consequently defined using a sphere of the 6 mm radius around the peak. As a result, the individual time series for each ROI was extracted from the significant voxels of the individual mask and summarized by the first eigenvariate. One participant was excluded as no voxels survived significance testing. Second, we specified three regressors of interest: genuine pain, pretended pain, and the video input condition (the combination of genuine pain and pretended pain). That we did not specify no-pain conditions was because (1) the pain conditions were our main focus, and (2) adding no-interest conditions would inevitably increase the model complexity. Then, a fully connected DCM model for each participant was created. Three parameters were specified: (1) bidirectional connections between regions and self-connections (matrix A), (2) modulatory effects (i.e., genuine pain and pretended pain) on the between-region connections (matrix B), and (3) driving inputs (i.e., the video input condition) into the model on both regions (matrix C) (*Zeidman et al., 2019a*). To remain parsimonious, we did not set modulatory effects on the self-connections in matrix A. Then the full DCM model was individually estimated. Finally, group-level DCM inference was performed using parametric empirical Bayes (*Zeidman et al., 2019b*). We conducted an automatic search over the entire model space (max. n = 256) using Bayesian model reduction and random-effects Bayesian model averaging, resulting in

a final group model that takes accuracy, complexity, and uncertainty into account (*Zeidman et al., 2019b*). The threshold of the Bayesian posterior probability was set to $p_p$ >0.95 (i.e., *strong evidence*) but we reported all parameters above $p_p$ >0.75 (i.e., *positive evidence*) for full transparency of the DCM results. Finally, a paired sample t-test was performed to compare modulatory effects between the genuine pain and the pretended pain conditions.

To probe whether task-related modulatory effects were associated with behavioral measurements, we performed multiple linear regression analyses of modulatory parameters with (1) the three behavioral ratings and (2) the empathy-related questionnaires (i.e., IRI, ECQ, and TAS). We set up regression models for the genuine pain condition and the pretended pain condition, respectively, in which the DCM parameters of modulatory effects were determined as dependent variables and the ratings of painful expressions in others, painful feelings in others, and unpleasantness in self as independent variables. Considering that interactions between behavioral ratings might contribute to the regression model, we tested five regression models (with and without interaction; See *Supplementary file 1*) for both genuine pain and pretended pain. Results showed that for both genuine pain and pretend pain, the model without any interaction outperformed other models. The results of the winning multiple regression model are reported in the Results section. We performed additional two regression models for both conditions in which DCM modulatory effects were set as dependent variables and scores of each subscale of all questionnaires were set as independent variables, respectively. Considering the number of independent variables was relatively large (>10), we performed the analyses for questionnaires using a stepwise regression approach. As two participants did not complete all three questionnaires, we excluded their data from the regression analyses. The statistical significance of the regression analysis was set to p<0.05. The multicollinearity for independent variables was diagnosed using the VIF that measures the correlation among independent variables, in the R package car (https://cran.r-project.org/web/packages/car/index.html). Here we used a rather conservative threshold of VIF < 5 as a sign of no severe multicollinearity (*Menard, 2002*; *James et al., 2013*).

## Acknowledgements

This work was supported by Chinese Scholarship Council (CSC) Grant (201604910515) and Vienna Doctoral School in Cognition, Behavior and Neuroscience (VDS CoBeNe) completion grant fellowship to YZ; the Vienna Science and Technology Fund (WWTF VRG13-007) to CL, and the Austrian Science Fund (FWF P 32686) to CL and MR. We thank Michael Schnödt, Lukas Repnik, Elisa Warmuth, Betty Geidel, Phan Ri, Sven Sander, Gvantsa Gogisvanidze, Robert Meyka, Anja Tritt, Tim Reinboth, Boryana Todorova, and Sophia Shea for help with data acquisition.

## Additional information

### Funding

| Funder | Grant reference number | Author |
|---|---|---|
| China Scholarship Council | 201604910515 | Yili Zhao |
| Vienna University of Technology | Dissertation Completion Fellowship | Yili Zhao |
| Vienna Science and Technology Fund | WWTF VRG13-007 | Claus Lamm |
| Austrian Science Fund | FWF [P32686] | Markus Rütgen Claus Lamm |

The funders had no role in study design, data collection and interpretation, or the decision to submit the work for publication.

### Author contributions

Yili Zhao, Conceptualization, Data curation, Formal analysis, Funding acquisition, Investigation, Methodology, Project administration, Resources, Software, Validation, Visualization, Writing - original draft, Writing - review and editing; Lei Zhang, Ronald Sladky, Conceptualization, Formal analysis,

Methodology, Resources, Software, Supervision, Validation, Writing - original draft, Writing - review and editing; Markus Rütgen, Conceptualization, Formal analysis, Funding acquisition, Methodology, Resources, Software, Supervision, Validation, Writing - original draft, Writing - review and editing; Claus Lamm, Conceptualization, Formal analysis, Funding acquisition, Methodology, Resources, Supervision, Validation, Writing - original draft, Writing - review and editing

### Author ORCIDs
Yili Zhao  http://orcid.org/0000-0003-4535-6636
Lei Zhang  http://orcid.org/0000-0002-9586-595X
Markus Rütgen  http://orcid.org/0000-0003-4947-7734
Ronald Sladky  http://orcid.org/0000-0002-5986-1572
Claus Lamm  http://orcid.org/0000-0002-5422-0653

### Ethics
Human subjects: Informed consent was obtained from all participants. For those participants whose images are to be published in eLife, consent to publish was obtained from each of them. More details can be found in Materials and Methods. The study was approved by the ethics committee of the Medical University of Vienna and was conducted in line with the latest version of the Declaration of Helsinki (2013).

### Decision letter and Author response
Decision letter https://doi.org/10.7554/eLife.69994.sa1
Author response https://doi.org/10.7554/eLife.69994.sa2

## Additional files

### Supplementary files
• Supplementary file 1. Model comparison of linear regression models with three behavioral ratings (independent variables) and the inhibitory effect (dependent variable) for genuine pain and pretended pain. Smaller AIC/BIC indicates better model fit. Results showed that M1 (without interaction; highlighted with underlining) was the best fitting model for both genuine pain and pretended pain.

• Transparent reporting form

### Data availability
All data needed to evaluate the conclusions in the paper are present in the paper. Raw functional imaging and behavioral data are deposited at https://doi.org/10.5281/zenodo.4783235. Processed behavioral data and individual DCM parameters are accessible at https://github.com/Yili-Zhao/Genuine_pretended-pain-task.git. Unthresholded statistical maps are available at https://identifiers.org/neurovault.collection:9949.

The following dataset was generated:

| Author(s) | Year | Dataset title | Dataset URL | Database and Identifier |
| --- | --- | --- | --- | --- |
| Zhao Y, Zhang L, Rütgen M, Sladky R, Lamm C | 2021 | The raw data of genuine and pretended pain task | https://zenodo.org/record/4783235#.YUC0C9NKgUE | Zenodo, 10.5281/zenodo.4783234 |

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
