## [Decision Letter]

**Acceptance summary:**

This is a novel exploration of the brain dynamics of empathy in response to facial expressions of simulated versus genuine pain in others. The findings suggest that activation of the anterior insula for genuine pain reflects affect sharing rather than automatic responses triggered by the perceptual salience of events. The paper will be of broad interest to an audience of social and affective neuroscientists interested in how humans track the emotional responses of others.

**Decision letter after peer review:**

Thank you for submitting your article "Neural dynamics between anterior insular cortex and right supramarginal gyrus dissociate genuine affect sharing from automatic responses to pretended pain" for consideration by *eLife*. Your article has been reviewed by 2 peer reviewers, one of whom is a member of our Board of Reviewing Editors, and the evaluation has been overseen by Christian Büchel as the Senior Editor. The reviewers have opted to remain anonymous.

Essential revisions:

1. It is not clear that the experimental paradigm allows one to unambiguously disentangle self vs. other processing (as mentioned in the abstract "..we investigated how affect sharing and self-other distinction interact.."). For example, genuine vs. pretended pain could be distinguished from the participants' own experience in a comparable way. The higher rating of unpleasantness for genuine pain in others does not necessarily mean that the participants cannot separate own from others' experiences. At the very least, this needs to be mentioned as a limitation of the study.

2. Similarly, the experimental design does not unambiguously allow the authors to disentangle genuine vs pretended pain from other factors, such as the differences in pain expression, painful feeling in others and higher unpleasantness in these two conditions. I understand that the intensity of pain expression, painful feeling in others and unpleasantness for others is inherently tied to genuine vs. pretended pain. But the authors already saw that the instruction of "genuine vs. pretend" influences ratings of pain expression. Hence, this allows two interpretations of the results: either the anterior Insula influence on the rSMG is driven by higher perceived pain expression, painful feelings in others and unpleasantness or by the conditions of genuine vs. pretended pain. Or (more likely) by an interaction between these factors. The authors need to consider the possibility of an interaction and discuss the different factors (pain expression, etc) that likely contribute to the differences between genuine vs. pretended pain.

3. It would help to explore the association between the aIns-rSMG interaction and pain expression ratings (or painful feeling in others or higher unpleasantness) in videos with genuine pain and pretended pain separately.

4. The multiple regression analyses revealed an association between the unpleasantness for the participants and the aIns, when accounting for the painful expression and the pain experienced by others. This, however, does not reveal the specificity of the aIns for encoding the unpleasantness for the participants. It might well be that variance is shared in the association between the aIns and pain expression and pain by the other and unpleasantness for the participants, but simply strongest for unpleasantness. Such ambiguity could be resolved by additional multiple regressions of (1) pain expression (controlling for pain by the other and unpleasantness for the participants) and (2) pain for the other (controlling for pain expression and unpleasantness for the participants). Further, it would be helpful to clarify if an intercept was included in the regression model. Finally, please provide a scatterplot of responses.

5. I would have liked more justification for focusing on the rSMG as an area implicated in self-other distinction given its role in selective attention (or perhaps they are related?). The inclusion of the rSMG into the DCM model was not clearly motivated. If this was based on previous literature, then one could also argue that the aMCC should be added into the model as well. Further, while the rSMG emerges as playing an important role, the current experiment does not reveal the actual processes driving this involvement. The authors state that the rSMG is involved in action observation or imitating emotions (page 9, line 200), however this is not substantiated by the current data.

6. In general, the results support the distinction between the experimental conditions of genuine vs. pretended pain in the aIns and a modulatory influence on the connectivity between the aIns and the rSMG. However, the authors aimed to test if genuine vs. pretended pain modulate regulatory influences from the aIns on the rSMG that are connected to self-other distinction (as proposed in the discussion page 8, line 170). Yet, any insights about self-other distinction are only inferred reversely, since there is no outcome that indicates how well participants distinguished between themselves and the other person. For example, in the discussion the authors state that: "we thus propose that the higher rSMG engagement in genuine pain conditions reflects an increasing demand for self-other distinction imposed by the stronger shared negative affect experiences in this condition". This is not supported by the current results.

7. Furthermore, the title mentions automated responses to pretended pain, which I could not understand, given the current results.

*Reviewer #1 (Recommendations for the authors):*

This is one of those rare studies where there is genuinely very little to critique. The authors have done an exemplary job in creating an innovative experimental paradigm with appropriate piloting, power calculations, and a very elegant experimental manipulation. Inclusion of targeted questionnaires to capture individual differences in empathic response is a further strength. I also appreciated the use of Dynamic Causal Modelling with Bayesian Model Reduction and random-effects Bayesian model averaging to understand the direction and magnitude of the aIns-SMG relationship. The manuscript itself is very well written and appropriate consideration is given to potential limitations and directions for future study.

*Reviewer #2 (Recommendations for the authors):*

Regression analysis: I would further like to ask the methodological detail, if an intercept was included in the regression model.

Regression analysis: please provide a scatterplot of responses.

---

## [Author Response]

Essential revisions:1. It is not clear that the experimental paradigm allows one to unambiguously disentangle self vs. other processing (as mentioned in the abstract "..we investigated how affect sharing and self-other distinction interact.."). For example, genuine vs. pretended pain could be distinguished from the participants' own experience in a comparable way. The higher rating of unpleasantness for genuine pain in others does not necessarily mean that the participants cannot separate own from others' experiences. At the very least, this needs to be mentioned as a limitation of the study.

We thank the reviewer for raising this issue and for prompting us to further clarify and better state our research purpose. In terms of its original theoretical foundation and motivation, the current study aimed to investigate whether and how neural signatures underlying two essential components of empathy, namely affect sharing and self-other distinction, track individual responses to genuine vs. pretended pain. We agree though that our experimental design does not allow to disentangle unequivocally the precise aspects of self- and other-related processing in the two main conditions of interest (genuine pain or pretended pain). We thus modified any wording suggesting otherwise, so as to avoid further misunderstanding by readers.

Accordingly, we have provided a more elaborate theoretical clarification in the Abstract and Introduction about our particular interest in studying self-other distinction and its neural correlates in the right supramarginal gyrus (rSMG) during empathy. We also mention as a potential limitation that our design did not aim to explicitly quantify self-other distinction.

In the manuscript, we have made the following changes:

1) We modified the sentence "[…] we investigated how affect sharing and self-other distinction interact […]" to:

“[…] we investigated how the brain network involved in affect sharing and self-other distinction underpinned […] ” in the Abstract (P. 1).

Besides, we modified another sentence “[…] to investigate the hypothesized distinct interactions between affective response and self-other distinction […]” to:

“[…] to investigate the hypothesized brain patterns of affective responses and self-other distinction […]” in the Introduction (P. 4).

2) We added sentences in the Discussion (P. 13):

“An additional limitation was that our study design did not aim to explicitly quantify self-other distinction. Rather, in line with previous research and based on our theoretical framework and rationale, we inferred the engagement of this process from the experimental conditions and the associated behavioral and neural responses. We expect our findings to prompt and inform future research designed to quantify and experimentally disentangle self- and other-related processes more explicitly.”

2. Similarly, the experimental design does not unambiguously allow the authors to disentangle genuine vs pretended pain from other factors, such as the differences in pain expression, painful feeling in others and higher unpleasantness in these two conditions. I understand that the intensity of pain expression, painful feeling in others and unpleasantness for others is inherently tied to genuine vs. pretended pain. But the authors already saw that the instruction of "genuine vs. pretend" influences ratings of pain expression. Hence, this allows two interpretations of the results: either the anterior Insula influence on the rSMG is driven by higher perceived pain expression, painful feelings in others and unpleasantness or by the conditions of genuine vs. pretended pain. Or (more likely) by an interaction between these factors. The authors need to consider the possibility of an interaction and discuss the different factors (pain expression, etc) that likely contribute to the differences between genuine vs. pretended pain.

We thank the reviewer for the thoughtful consideration of different factors that might contribute to disentangle genuine pain vs. pretend pain. One thing we would like to address beforehand: to disentangle the specific contributors underlying the manipulation is not the main focus for the current study, as (1) our primary aim was to study the effects of the experimental manipulation as a whole; we thus used the three behavioral ratings mainly to collect additional information on and to interpret the expected effects of the manipulation, and (2) these factors (and their behavioral measures) are inherently (cor)related and hard to be disentangled precisely anyways, as mentioned by the reviewer and as shown by extensive previous research both by our and other groups.

Nonetheless, in the revised manuscript, we have now:

(1) Discussed how different factors possibly interact and in this way contribute to the differences (in the modulatory effect) between genuine vs. pretended pain, in the Discussion (P. 11):

“We speculate that a dynamic interaction between sensory-driven and control processes is underlying the modulatory effect: when individuals realized after an initial sensory-driven response to the facial expression that it was not genuinely expressing pain, control and appraisal processes led to a reappraisal of the triggered emotional response, and thus a dampening of the unpleasantness.”

(2) Performed additional linear regression models and model comparison (see details in the response to comment #3) to investigate whether an interaction between behavioral measures could be a potential contributor to the modulatory effect of genuine pain and pretended pain; in short, the model without interactions is the winning model both for genuine pain and pretended pain.

We have now discussed this result (P. 11):

“Model comparison showed that the best model to explain the inhibitory effect with the behavioral ratings for both the genuine and pretended pain is the model without interactions between ratings. That is, if any behavioral rating contributed to the modulation of aIns to rSMG, the effect would be more likely coming from single ratings rather than their interactions. Specifically, we found […]”

3. It would help to explore the association between the aIns-rSMG interaction and pain expression ratings (or painful feeling in others or higher unpleasantness) in videos with genuine pain and pretended pain separately.

We thank the reviewer for this suggestion for further analysis.

We performed additional linear regression models (with and without interaction) and model comparison to explore whether any interaction between behavioral ratings heavily contributed to the modulatory effect. Results showed that the model without interaction was the most efficient model for both conditions.

We report the additional analyses as follows:

in the Methods section (P. 24-25):

“Considering that interactions between behavioral ratings might contribute to the regression model, we tested five regression models (with and without interaction; see Supplementary Table 1) for both genuine pain and pretended pain. Results showed that for both genuine pain and pretend pain, the model without any interaction outperformed other models.”

Supplementary Table 1. Model comparison of linear regression models with three behavioral ratings (independent variables) and the inhibitory effect (dependent variable) for genuine pain and pretended pain. Smaller AIC/BIC indicates better model fit. Results showed that M1 (without interaction; highlighted with underlining) was the best fitting model for both genuine pain and pretended pain.

Accordingly, we now report the results of the winning model of the multiple regression analyses, instead of the original stepwise regression. These analyses found that only the rating of painful feelings in others was significant for genuine pain, while no significant effects whatsoever were found for pretended pain.

In the manuscript, we have made the following changes:

1) We modified “stepwise linear regression” to “multiple linear regression” in the Methods, Results, and Figure 3 legend (P. 24, P. 7, and P. 37).

(2) We added the sentence “The results of the winning multiple regression model are reported in the Results section.” in the Methods (P. 25).

(3) We added the results of the multiple regression analyses for genuine pain and pretended pain, in the Results section (P. 7-8):

“For the genuine pain condition, we find that the modulatory effect was significantly related to the rating of painful feelings in others (t = 2.317, p = 0.026) but not related to the rating of either painful expressions in others (t = -1.492, p = 0.144) or unpleasantness in self (t = 0.058, p = 0.954). For the pretended pain condition, none of the ratings was significantly related to the modulatory effect (Figure 3D).”

4. The multiple regression analyses revealed an association between the unpleasantness for the participants and the aIns, when accounting for the painful expression and the pain experienced by others. This, however, does not reveal the specificity of the aIns for encoding the unpleasantness for the participants. It might well be that variance is shared in the association between the aIns and pain expression and pain by the other and unpleasantness for the participants, but simply strongest for unpleasantness. Such ambiguity could be resolved by additional multiple regressions of (1) pain expression (controlling for pain by the other and unpleasantness for the participants) and (2) pain for the other (controlling for pain expression and unpleasantness for the participants). Further, it would be helpful to clarify if an intercept was included in the regression model. Finally, please provide a scatterplot of responses.

We thank the reviewer for this comment. As an overall premise, please note that we would not want to claim that the aIns is specifically engaged in encoding affective activities without any engagement of other processes; instead, we are entirely aware that the aIns activation participates in a variety of affective and cognitive processes. Nonetheless, our original multiple regression models were performed as a second-level group analysis with all three ratings as independent variables. Results showed that only the rating of “unpleasantness in self” was significant, rather than all ratings that were universally influenced by domain-general factors.

As the reviewer suggested, we additionally performed five multiple regression analyses with all possible orders of three behavioral measures to test whether the order matters. In the end, we found consistent results across all six regression analyses, suggesting that the selective correlation of aIns and the rating of unpleasantness in self was robust.

In the manuscript, we have:

1) Modified “specifically” to “selectively” in the Results (P. 6).

(2) Added the content in the Methods (P. 22) “To test whether the order of entering ratings into the regression model influence the results, we performed five additional regression analyses with all possible orders of three ratings. The results were consistent across all six regression models, and we only showed the result for one regression (i.e., expression + feeling + unpleasantness) in the Results section.”

(3) Modified the sentence in the Results (P. 6) “We found significant clusters in bilateral aIns, visual cortex, and cerebellum (Figure 2B); notably, when statistically accounting for ratings of painful expressions in others and painful feelings in others, all three clusters were exclusively explained by the ratings of self-unpleasantness.” to:

“We found significant clusters in bilateral aIns, visual cortex, and cerebellum that could be selectively explained by the ratings of self-unpleasantness and could not be explained by either the ratings of painful expressions in others or painful feelings in others (Figure 2B).”

(4) Modified the sentence in the Discussion (P. 9) “[…] but the increased activation in aIns was also selectively correlated with ratings of self-oriented unpleasantness (i.e., after statistically accounting for painful expressions and painful feelings in others) […]” to:

“[…] but the increased activation in aIns was also selectively correlated with ratings of self-oriented unpleasantness and was not correlated with neither other-related painful expressions nor painful feelings in terms of the regression analysis […]”

and added the sentence “[…] (otherwise the increased aIns activation should also be explained by other behavioral ratings in the sense of shared influence by domain-general effects).”

(5) Modified the legend for Figure 3 (P. 37) “[…] revealed a positive correlation between the inhibitory effect and painful feelings in others (after accounting for the other two ratings) for genuine pain […]” to:

“[…] revealed a positive correlation between the inhibitory effect and painful feelings in others and not with other two ratings for genuine pain […]”

For the intercept, yes, the intercept was included in the multiple regression model. We added the following sentence in the Methods (P. 22):

“An intercept was added in the model.”

For the scatterplot of responses, note that we now show two scatterplots of the left and right aIns in Figure 2B (P. 35).

“[…] that were positively correlated with the ratings of unpleasantness in self comparing genuine pain vs. pretended pain. […] The lines of the scatterplots represent the fitted regression lines, bands indicate a 95% CI.”

5. I would have liked more justification for focusing on the rSMG as an area implicated in self-other distinction given its role in selective attention (or perhaps they are related?). The inclusion of the rSMG into the DCM model was not clearly motivated. If this was based on previous literature, then one could also argue that the aMCC should be added into the model as well. Further, while the rSMG emerges as playing an important role, the current experiment does not reveal the actual processes driving this involvement. The authors state that the rSMG is involved in action observation or imitating emotions (page 9, line 200), however this is not substantiated by the current data.

We appreciate the reviewer’s comment that shows we seemed not to convey clearly why we have postulated a role of rSMG. We have now made our rationale more explicit and clear.

We have now:

1) Modified the clarification of rSMG in the Discussion (P. 10):

“The inferior parietal lobule was shown to be generally engaged in selective attention, action observation and imitating emotions (Bach et al., 2010; Pokorny et al., 2015; Gola et al., 2017; Hawco et al., 2017). Importantly, a specific role in affective rather than cognitive self-other distinction has been consistently identified for rSMG (Silani et al., 2013; Steinbeis et al., 2015; Bukowski et al., 2020). […]”

2) Added further clarification in the Discussion (P. 12) after the sentence “ […] the correlation findings provide further evidence that the modulation of aIns to rSMG is implicated in encoding others’ emotional states,” with:

“which serves as a functional foundation for self-other processing […] This regulation cannot be totally attributed to domain-general processes, otherwise other ratings should have also explained this variation.”

Additionally, we agree re: aMCC, which we also predicted to play a role; but it was not the case at least in our data. In fact, we have already addressed this in the original version of the manuscript. (maintained on P. 7 of revised manuscript.): “Our original analysis plan was to include aMCC in the DCM analyses, but based on the fact that aMCC did not show as strong evidence (in terms of the multiple regression analysis) as the aIns of being involved in our task, we decided to use a more parsimonious DCM model without the aMCC.”

6. In general, the results support the distinction between the experimental conditions of genuine vs. pretended pain in the aIns and a modulatory influence on the connectivity between the aIns and the rSMG. However, the authors aimed to test if genuine vs. pretended pain modulate regulatory influences from the aIns on the rSMG that are connected to self-other distinction (as proposed in the discussion page 8, line 170). Yet, any insights about self-other distinction are only inferred reversely, since there is no outcome that indicates how well participants distinguished between themselves and the other person. For example, in the discussion the authors state that: "we thus propose that the higher rSMG engagement in genuine pain conditions reflects an increasing demand for self-other distinction imposed by the stronger shared negative affect experiences in this condition". This is not supported by the current results.

We thank the reviewer for this comment, which somewhat follows up on similar arguments made and replied to in comment #1 above. Indeed, we fully agree that our design did not allow us to quantify self-other distinction, but that we inferred its engagement based on a strong theoretical motivation and the replication of previous findings on rSMG involvement during self-other distinction. As outlined above (cf. #1), this limitation was added to the revised manuscript.

We also adjusted the way of reasoning for which we put the theoretical explanation ahead of the inference so that readers can better realize this statement is supported by stronger theoretical motivation in the Discussion (P. 10):

First “Theoretical models of empathy […]” and then “Concerning the current finding, we thus propose that […]”

7. Furthermore, the title mentions automated responses to pretended pain, which I could not understand, given the current results.

We thank the reviewer for pointing out the potential ambiguity in the title. We agree it may be somewhat “imprecise”, and have revised the title accordingly (P. 1):

“Neural dynamics between anterior insular cortex and right supramarginal gyrus dissociate genuine affect sharing from perceptual saliency of pretended pain”.

Reviewer #1 (Recommendations for the authors):This is one of those rare studies where there is genuinely very little to critique. The authors have done an exemplary job in creating an innovative experimental paradigm with appropriate piloting, power calculations, and a very elegant experimental manipulation. Inclusion of targeted questionnaires to capture individual differences in empathic response is a further strength. I also appreciated the use of Dynamic Causal Modelling with Bayesian Model Reduction and random-effects Bayesian model averaging to understand the direction and magnitude of the aIns-SMG relationship. The manuscript itself is very well written and appropriate consideration is given to potential limitations and directions for future study.

Reviewer #2 (Recommendations for the authors):Regression analysis: I would further like to ask the methodological detail, if an intercept was included in the regression model.

Our public reply to this comment will be identical as the one to the main review comment #4 above, and thus as follows:

We thank the reviewer for raising this question. For the intercept, yes, the intercept was included in the multiple regression model. We added the following sentence in the Methods (P. 22):

“An intercept was added in the model.”

Regression analysis: please provide a scatterplot of responses.

We thank the reviewer for this suggestion. For the scatterplot of responses, note that we now show two scatterplots of the left and right aIns in Figure 2B (P. 35).

“[…] that were positively correlated with the ratings of unpleasantness in self comparing genuine pain vs. pretended pain. […] The lines of the scatterplots represent the fitted regression lines, bands indicate a 95% CI.”